



# Barotropic Tides in MPAS-Ocean: Impact of Ice Shelf Cavities

Nairita Pal[1], Kristin N. Barton[2], Mark R. Petersen[1], Steven R. Brus[3], Darren Engwirda[1], Brian K. Arbic[2], Andrew F. Roberts[1], Joannes J. Westerink[4], and Damrongsak Wirasaet[4]

[1]Los Alamos National Laboratory, Los Alamos, New Mexico 87545, USA
[2]University of Michigan, Ann Arbor, MI, USA
[3]Argonne National Laboratory, Lemont IL 60439, USA
[4]University of Notre Dame, 156 Fitzpatrick Hall, Notre Dame, IN, USA

**Correspondence:** Nairita Pal (nairitap2009@gmail.com)

**Abstract.** Oceanic tides are seldom represented in Earth System Models (ESMs) owing to the need for high horizontal resolution to accurately represent the associated barotropic waves close to coasts. This paper presents results of tides implemented in the Model for Prediction Across Scales–Ocean or MPAS-Ocean, which is the ocean component within the US Department of Energy developed Energy Exascale Earth System Model (E3SM). MPAS-Ocean circumvents the limitation of low resolution using unstructured global meshing. We are at this stage simulating the largest semidiurnal (M2, S2, N2) and diurnal (K1, O1) tidal constituents in a single layer version of MPAS-O. First, we show that the tidal constituents calculated using MPAS-Ocean closely agree with TPXO8 results when suitably tuned topographic wave drag and bottom drag coefficients are employed. Thereafter, we present the sensitivity of global tidal evolution due to the presence of Antarctic ice shelf cavities. The effect of ice shelves on the amplitude and phase of tidal constituents are presented. Lower values of complex errors (with respect to TPX08 results) for the M2 tidal constituents are observed when ice shelf is added in the simulations, with particularly strong improvement in the Southern Ocean. Our work points towards future research with varying Antarctic ice shelf geometries and sea ice coupling that might lead to better comparison and prediction of tides, and thus better prediction of sea-level rise and also the future climate variability.

## 1 Introduction

Ocean tides are generated due to the gravitational force from celestial bodies—the Sun, Moon and the Earth. Such gravitational dynamics causes the sea surface height of the global ocean to oscillate at precise amplitude and frequency (Mellor, 1996). These oscillations are dominated by diurnal (daily) and semidiurnal (twice daily) constituents or frequencies. The tide plays an important role in a broad range of global oceanic processes including oceanic mixing and heat, salt and BGC fluxes (Vic et al., 2019; Ledwell et al., 2000; Munk, 1997), sedimentation (Dalrymple and James, 2010), and biological processes and resulting habitats (Knox et al., 2018). Oceanic mixing due to tides, winds and waves contributes to redistribution of heat from the equator to the poles. Thus understanding and predicting tides and tidal mixing is essential to predict global climate variability. Accurate representation of tides are necessary, not only to predict ocean currents and future climate variability, but also to remove tidal





noise during satellite altimetry or satellite gravimetry (Coleman, 2001; Zaron and DeCarvalho, 2016) since tidal signals affect different kinds of observational data, ranging from space geodetic observations to measurement of ocean currents.

25  In this paper, we focus on the mechanical interactions of the ocean tides with the Antarctic ice shelf cavities. Needless to say, accurately representing such tidal interactions with ice shelves are indispensable in the current scenario of a warming climate, melting ice shelves, and constantly evolving ice shelf geometries. Ice shelves are permanent floating sheets of ice that are connected or "grounded" to a landmass (see Fig. 1). Most of the world's ice shelves hug the coast of Antarctica. However, ice shelves can also form wherever ice flows from land into cold ocean waters, including some glaciers in the Northern Hemisphere.

30 The point at which the ice shelf loses contact with the bed is called the "grounding line". This grounding line can migrate with the tide, due to the elastic interactions between the tides and ice or the tidal flexure on the surface of an ice shelf (Brunt et al., 2010; Padman et al., 2018). The region over which the ice shelf loses contact with the bedrock and just floats in open ocean is called the "grounding zone", and in this region the ice shelves directly interact with oceans, and thus ocean tides. There are a number of detailed works on ocean/ice shelf interactions (Jacobs et al., 1992; Straneo et al., 2013; Warburton et al.,

35 2020; Turner et al., 2017). In this work, we present the impacts of the Antarctic ice shelf cavities on the dynamics of global tides, simulated using the LANL–led global ocean model known as "Model for Prediction Across Scales – Ocean" or MPAS-Ocean (Ringler et al., 2013; Petersen et al., 2015). We have modified MPAS-Ocean to work in a barotropic or "single-layer" framework to increase computational efficiency.

  The barotropic, or a single-layer, global tides model within MPAS–Ocean is supplemented with specific schemes to dissipate

40 tidal energy. The total energy of the tidal motion in the oceans across the world is about 3.5 TW (Egbert and Ray, 2003), which was originally believed to be dissipated only through bottom friction in shallow coastal shelf seas (Taylor, 1920). Thus, in the early barotropic tide models the tidal dissipation term was either linear or quadratic bottom drag in the coastal oceans, with the drag coefficients appropriately tuned to match observations (Schwiderski, 1980; Le Provost et al., 1994). However, works of Egbert and Ray (2000) showed that about a third of the barotropic tidal dissipation occurs in the deep ocean, thus

45 debunking previous assumptions that all tidal energy is dissipated in shallow water. Works of Munk (1966) and Munk and Wunsch (1998) established that baroclinic tides contribute strongly to tidal energy dissipation. Tidal currents flowing over topography in a stratified ocean can give rise to tidal-period oscillations, known as internal or baroclinic tides. In fact, energy is transferred from the barotropic to the baroclinic tide at rough topography (Munk and Wunsch, 1998). The energy conversion from barotropic to the baroclinic tide at rough topography is represented through a linear wave drag term (Stigebrandt, 1999;

50 Jayne and St. Laurent, 2001). Such wave drags are commonly known as "internal wave drag" in literature (Egbert et al., 2004; Arbic et al., 2004), and have been shown to improve tidal elevations by reducing root-mean-square elevation error with observations (Egbert et al., 2004; Arbic et al., 2004; Green and Nycander, 2013; Lyard et al., 2006). The internal wave drag scheme can be implemented with or without a tunable parameter. In our work, which is an early effort at introducing a linear wave drag scheme in a tide model implemented in MPAS-Ocean, we use a scalar wave drag formulation with a tunable

55 parameter. Our scheme, originally implemented by Jayne and St. Laurent (2001), is based on a linear scaling relationship, and strongly depends on a tunable parameter. As we show later, our tuning parameter essentially sets the length scale associated with dissipation.





Some drag schemes (Egbert et al., 2004) are derived using linear transformations of the bottom topography (Bell Jr., 1975) and do not have a tunable parameter. Such schemes have been shown to perform better when applied to barotropic models,

with global dissipation rates close to TPXO7.2 (Green and Nycander, 2013). However, the drawback of schemes without a tunable parameter is that the dissipation rates and root-mean-square errors strongly depend on the global bathymetry and stratification databases, and thus do not guarantee an optimal prediction of elevation and dissipation rates. For example, bathymetric databases might not resolve all abyssal features. The analytically computed Nycander scheme has been shown to increase regional and global dissipation rates due to the inclusion of abyssal hill roughness on ocean spreading ridges (Melet et al.,

2013). On the other hand, high resolution bathymetric data bases increase the linear wave drag strength indicating a need for more tuning (Nycander, 2005; Zilberman et al., 2009). The reason for such increase in wave drag has been attributed to a supercritical topography (Nikurashin and Ferrari, 2011; Scott et al., 2011), but a tunable parameter is still needed to correct the effect.

The particular scalar scheme that we utilize in this study is the Jayne and St. Laurent (2001) scheme on a barotropic global

ocean model forced with five major tidal constituents: M2, S2, N2, K1, and O1. We show that our model performs very well with respect to root-mean-square tidal elevation comparisons against TPXO8 data. We present a sensitivity study using meshes of varying resolution, and show that the root-mean-square error values improve when we use high-resolution meshes. In the last section of the paper, we focus on the mechanical interactions of the ocean tides with the Antarctic ice shelf cavities, and show that inclusion of ice shelf cavities improve the tidal elevation comparison results against TPXO8 data.

One other important factor that must be accounted for in global tide models is the self-attraction and loading (SAL). SAL accounts for a combination of effects: the deformation of the Earth's crust due to mass loading and the self-gravitation of the load-deformed Earth as well as of the ocean tide itself (Hendershott, 1972). Self-attraction and loading can change tidal amplitudes to first-order, up to 20% in some regions, and also significantly impacts tidal phases and amphidromic points (Gordeev et al., 1977). Full calculation of SAL calls for convolution of tidal elevation with a proper Green's function or

a multiplication with load Love numbers in the spectral—i.e., spherical harmonic—domain Ray (1998). However, due to computational efficiency, we have used a scalar SAL term in the major part of this study (Accad and Pekeris, 1978). Full calculation of SAL has been recently investigated in detail in Barton et al. (submitted) and Brus et al. (submitted).

The paper is organized as follows. In Section 2, we present a brief overview of our model components, with focus on new modifications and developments within the MPAS-Ocean framework. In Section 3, we show some verification and validation re-

sults for our barotropic single-layer MPAS-Ocean ocean framework against analytical solutions from a planar two–dimensional ocean test case. We show that our barotropic ocean model reproduces known theoretical results. In Section 4, we present some background on TPXO8 data and the MPAS-Ocean mesh used for the simulations. In Section 5, we present the MPAS-Ocean model's sea surface elevation root-mean-square error results against observational TPXO8 data, the sensitivity of the results to different meshes, the impact of adding Antarctic ice shelf cavities, and the effect of adding a full (inline) self attraction and

loading term in our simulations. We finish our paper with discussions, conclusions and future directions in Section 6.





## 2   Model Formulation and Equations

This work is a systematic analysis of tides implemented in MPAS-Ocean, including the addition of ice shelf cavities. MPAS-Ocean is the ocean component of the DOE's Energy Exascale Earth System Model (E3SM) (Golaz et al., 2019; Petersen et al., 2019). MPAS-Ocean is solved using finite volume methods (FVM) on an unstructured grid, and this is where our work has an extra edge. The MPAS-Ocean code has the ability to run on unstructured meshes, which helps resolve sharp bathymetry critical for tides. This multi-resolution approach is built upon two key components: a variable-resolution mesh with exceptional mesh-quality characteristics; and a finite–volume method that maintains all of its conservation properties even when implemented on a highly nonuniform grid. The unstructured variable–resolution mesh is required to resolve coastal processes in greater detail, while maintaining a relatively low resolution for the open ocean processes which are relatively simpler to parameterize. Such variable resolution is achieved using the Spherical Centroidal Voronoi Tessellations or SCVTs (Ringler et al., 2008). Numerical algorithms specifically designed for these grids guarantee that volume and energy are conserved (Ringler et al., 2010). The time-stepping scheme is the fourth-order Runge-Kutta method.

We present results of global tidal evolution over a number of horizontal mesh configurations, ranging from uniform 64 km to 8km configurations as well as variable-resolution designs. Such single-layer framework is ideal for a seamless implementation and testing of new features (e.g., the topographic wave drag and bottom friction schemes that we use). The governing momentum equations for MPAS-O that we use in our study are the shallow–water equations, written in so-called vector-invariant form (Ringler et al., 2010):

$$\frac{\partial \mathbf{u}}{\partial t} + (\nabla \times \mathbf{u} + f\mathbf{k}) \times \mathbf{u} \quad = \quad -\nabla K - g\nabla(\eta - \eta_{SAL} - \eta_{EQ}) - \frac{\nabla p^s}{\rho_0} - \chi\frac{\mathcal{C}\mathbf{u}}{H} - \frac{\mathcal{C}_D|\mathbf{u}|\mathbf{u}}{H}, \tag{1}$$

$$\frac{\partial h}{\partial t} + \nabla \cdot (h\mathbf{u}) \quad = \quad 0, \tag{2}$$

$$p(x,y,z) \quad = \quad p^s(x,y) + \rho_0 g h, \tag{3}$$

where $\mathbf{u}$ represents the depth-averaged horizontal velocity, $t$ is the time coordinate, $f$ is the Coriolis parameter, $\mathbf{k}$ is the vertical unit vector, $K = |\mathbf{u}|^2/2$ is the kinetic energy, $g$ is the gravitational acceleration constant, $\eta$ is the sea-surface height relative to the ocean surface, $\eta_{EQ}$ is the equilibrium tide, $\eta_{SAL}$ is the perturbation of tidal elevations due to SAL, $\chi$ is a tunable scalar dimensionless wave drag coefficient, $\frac{\mathcal{C}}{h}$ is a topographic wave drag time scale, $H$ is the resting depth of the ocean, $\mathcal{C}_D$ is the bottom friction coefficient, and $h$ is the total ocean thickness such that $H + \eta = h$. The full form of the drag terms in (1) would use the total thickness $h$, but our implementation uses the linearized version with the resting depth $H$. For the linear drag scheme, we use the formulation due to Jayne and St. Laurent (2001), with the inverse timescale $\frac{H}{\mathcal{C}}$ computed according to Buijsman et al. (2016). Further details are provided in Section 2.3. Here $p^s$ represents the surface pressure on the ocean due to the floating ice shelf cavities, (see Section 2.5), and $p$ refers to the total pressure on the vertical column of water.

We have the following important modification to MPAS-Ocean in our model: (a) we introduce an external forcing due to astronomical tides (Barton et al., submitted; Brus et al., submitted); (b) we introduce a new drag scheme due to friction with





the sea bed representing the log-law of wall turbulence; and a new drag scheme called topographic wave drag due to scattering with rough topography; (c) dynamics due to Antarctic ice shelf cavities, together with a full inline SAL.

## 2.1 Tidal forcing

The tidal forcing is obtained from an astronomical potential among celestial bodies (see Barton et al. (submitted) for details). Here we just state the governing equations for the sea surface height perturbation due to tides. The perturbation to the sea surface height $\eta_{EQ}$ which is a combination of principal semi-diurnal and diurnal tides.

$$\eta_{EQsd,c} = A_c f_c(t_{ref}) L \cos^2(\phi) \cos[\omega_c(t - t_{ref}) + \chi_c(t_{ref}) + \nu_c(t_{ref}) + 2\lambda] \tag{4}$$

$$\eta_{EQd,c} = A_c f_c(t_{ref}) L \sin(2\phi) \cos[\omega_c(t - t_{ref}) + \chi_c(t_{ref}) + \nu_c(t_{ref}) + \lambda], \tag{5}$$

where these terms are valid for semidiurnal ($sd$) and diurnal ($d$) tidal constituents (Arbic et al., 2018). The total forcing comes from summing over each of the constituents, $c$. Here $A$ and $\omega$ are the forcing amplitude and frequency, respectively, dependent on the tidal constituent, $t_{ref}$ is a specified reference time, $t$ is time, $\phi$ is latitude, $\lambda$ is longitude, $\chi(t_{ref})$ is an astronomical argument dependent on tidal constituent, and $f(t_{ref})$ and $\nu(t_{ref})$ are amplitude and phase nodal factors accounting for small known astronomical modulations in the tidal forcing. $L = 1 + k_2 - h_2$ represents the body tide Love numbers account for

changes in the gravitational potential ($k_2$) due to deformation of the Earth's crust and mantle from tidal forcing ($h_2$).

## 2.2 Drag based on Ocean Bottom Roughness

An ocean bottom–depth–based friction coefficient is important when we want to use a smaller value in deep water (for non-tidal energetics tuning, say) and a high value for tides in shallow regions. We implement a bottom friction coefficient based on the ocean bottom roughness, in the MPAS-Ocean model code. The friction coefficient $C_D$ in Eq. (1) is evaluated according to

the following formula (Oey, 2006)

$$C_D = \max\left[\frac{\kappa^2}{\ln\left[\frac{z_b}{z_0}\right]}, C_{d,\min}\right] \tag{6}$$

where $\kappa = 0.4$ is the von Karman constant (Von Kármán, 1931), $z_0$ is the roughness parameter and $z_b$ is the ocean layer thickness (i.e., the sea surface height + the ocean bottom depth). $C_{d,\min}$ is the minimum quadratic bottom friction coefficient. Typically, in previous studies, $z_0 = 10mm$, and $C_{d,\min} = 0.0025$. The derivation of $C_d$ is given in (Oey, 2006) and is obtained

by matching the near–bottom modeled velocity to the Law of the Wall (Schlichting and Gersten, 2016).

## 2.3 Topographic Wave Drag

As we discuss in the Introduction, we utilize the Jayne and St. Laurent (2001) scheme to calculate the topographic wave drag in this study. This is a scalar scheme, in which the wave drag strength is independent of the flow direction. Works by Egbert et al. (2004) show that the scalar scheme produces reasonable agreement with observations after being appropriately tuned. This

scalar scheme is easier to implement than the more sophisticated tensor–based scheme Nycander (2005). In the tensor–based





scheme, the tensor components are functions of the directionality of topographic roughness, and hence, the wave drag strength depends on the flow direction relative to the rough topography.

The term $\chi\dfrac{\mathcal{C}\mathbf{u}}{H}$ represents the topographic wave drag on the velocity $\mathbf{u}$. The factor $\dfrac{\mathcal{C}}{H}$ is an inverse time scale and $\chi$ is a tunable drag coefficient. As we discussed in the introduction, this is a scalar linear wave drag scheme, or the Jayne and St.

Laurent formulation for wave drag (Jayne and St. Laurent, 2001; Buijsman et al., 2015)

$$\mathcal{C} = \frac{\pi}{L}\hat{H}^2 N_b \tag{7}$$

Here $\hat{H}$ is the bottom roughness, $N_b$ is the buoyancy frequency at the bottom, and $L$ is the wavelength of the bottom topography typically set in $10km$ (Jayne and St. Laurent, 2001; Buijsman et al., 2015). We implement the wave drag term as $\chi\mathcal{C}\mathbf{u}$ representing a linear drag on the velocity $\mathbf{u}$. The factor $\mathcal{C}$ $(= \hat{H}^2 N_b)$ is a data field we obtain following the procedure described

in Buijsman et al. (2015) and $\chi$ is a tunable drag coefficient. It is interesting to note here that after tuning, the optimum value of $\chi$ settles at a value between $1.08$ and $1.44$ for the high-resolution meshes, which is in the same range as the pre-factor $\dfrac{\pi}{L}$ in Eq. (7).

### 2.4 Self Attraction and Loading (SAL)

In the first part of the work, SAL is implemented in MPAS-Ocean via the scalar approximation (Accad and Pekeris, 1978; Ray,
165 1998)

$$\eta_{SAL} = \beta\eta, \tag{8}$$

where $\eta$ is the sea-surface height prior to alterations, and $\beta = 0.09$ is a scalar parameter used to approximate the influence of SAL. This approximation is a computationally inexpensive method that is sufficiently accurate for many cases. However, this does not capture the spatial dependence and large-scale smoothing of the full calculation.

To capture the full spatial dependence, in the latter part of the paper, SAL is implemented as additional body force via the sea surface height gradient term in Eq. (1). As detailed in Barton et al. (submitted), we express the inline SAL for tides in terms of the spherical harmonic decomposition of the sea surface height (Hendershott, 1972)

$$\eta_{SAL} = \sum_n \frac{3\rho_0}{\rho_{earth}(2n+1)}(1 + k'_n - h'_n)\eta_n. \tag{9}$$

where each spherical harmonic sea surface height term $\eta_n$ is multiplied by a scalar coefficient (Wang et al., 2012). Here $\rho_0$
is the average density of seawater, $\rho_{earth}$ is the average density of the solid earth, and the multiplicative term $(1 + k'_n - h'_n)$ represents load Love numbers corresponding to physical effects of SAL. The "1", $k'_n$, and $h'_n$ terms account for gravitational self-attraction of the ocean, gravitational self-attraction of the deformed solid earth, and deformation due to loading of the solid earth respectively. However, the usage of sea-surface height for calculating SAL is only appropriate for tides and wind-driven barotropic motions. For other motions one must use bottom pressure anomalies.





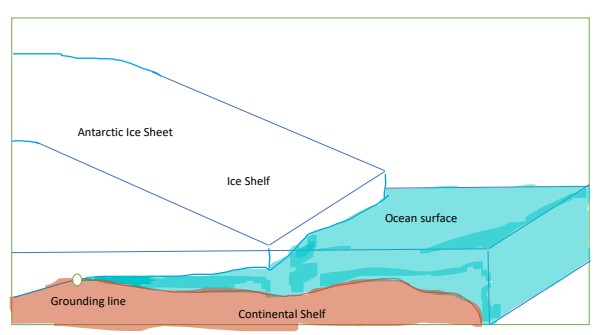

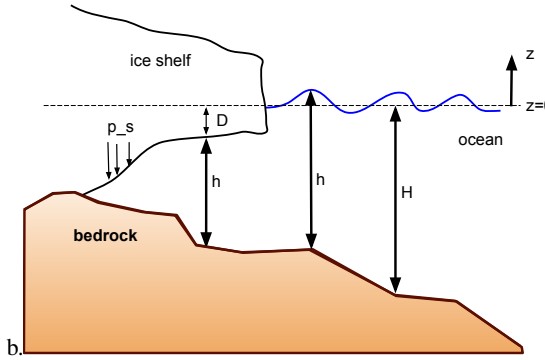

**Figure 1.** Schematic of ice shelf cavity. The grounding line (between the ice shelf and continental shelf) is marked with a white circle (b) Schematic showing the dynamic water thickness $h$, reference thickness $H$, ice draft $D$ and surface pressure $p_s$.

## 2.5 Ice shelf Cavities and their connection to pressure

Within MPAS-Ocean the ocean domain is extended to include cavities under Antarctic ice shelves (Fig. 1, Jeong et al. (2020); Comeau et al. (2022)). In our work, we do not study the dynamic ice sheet/ocean coupling, so the ice shelf/ocean interface is essentially static in time, adjusting only to the relatively small changes in dynamic pressure from the ocean. We use a constant density, salinity and temperature in our study, and thus do not evolve these tracers separately. Since we have a single vertical ocean layer, we use a hydrostatic formulation for pressure with a linear dependency on the height of the vertical column (see Eq. (3)). The implementation of ice shelf cavities require introducing an extra pressure term to Eq. (1) to account for the depression of sea–surface height below the Antarctic ice sheet and above the continental shelf (see Fig. 1). This extra pressure $p^s$ represents the weight of floating ice-shelves. The vertical coordinate below ice shelves in multi-layer MPAS-O is described in Comeau et al. (2022). For the single-layer version, the pressure is simply computed as $p^s = \rho g D$ where $D$ is the ice shelf draft below the open-ocean sea surface datum, and $\rho$ is the density of the displaced water (Fig **??**b).

## 3 Idealized Experiments

In this section we show that Eqs. (1)–(3), with the tidal forcing set to zero, reproduce known theoretical results for the ideal test cases: (a) Aquaplanet gravity wave test case and (b) Stommel's solution on a planar ocean basin. These two test cases are used to verify to operation of the single-layer shallow water equations on the sphere, including the spatial discretization of the advection, Coriolis, and pressure gradient terms, and the time-stepping scheme.

An aquaplanet is a planet without any continents. If initialized with a Gaussian hump sea-surface height test case, the Gaussian hump spreads at the speed of gravity waves in the ocean. The speed of gravity waves would be $c = \sqrt{gH}$, where $g$ is the acceleration due to gravity $= 9.8 ms^{-2}$, $H$ is the bottom depth of the ocean (here $1000m$). We have verified that our MPAS-Ocean simulations indeed produce a gravity wave speed of $c = \sqrt{gH}$ (details in Section A in the Appendix).



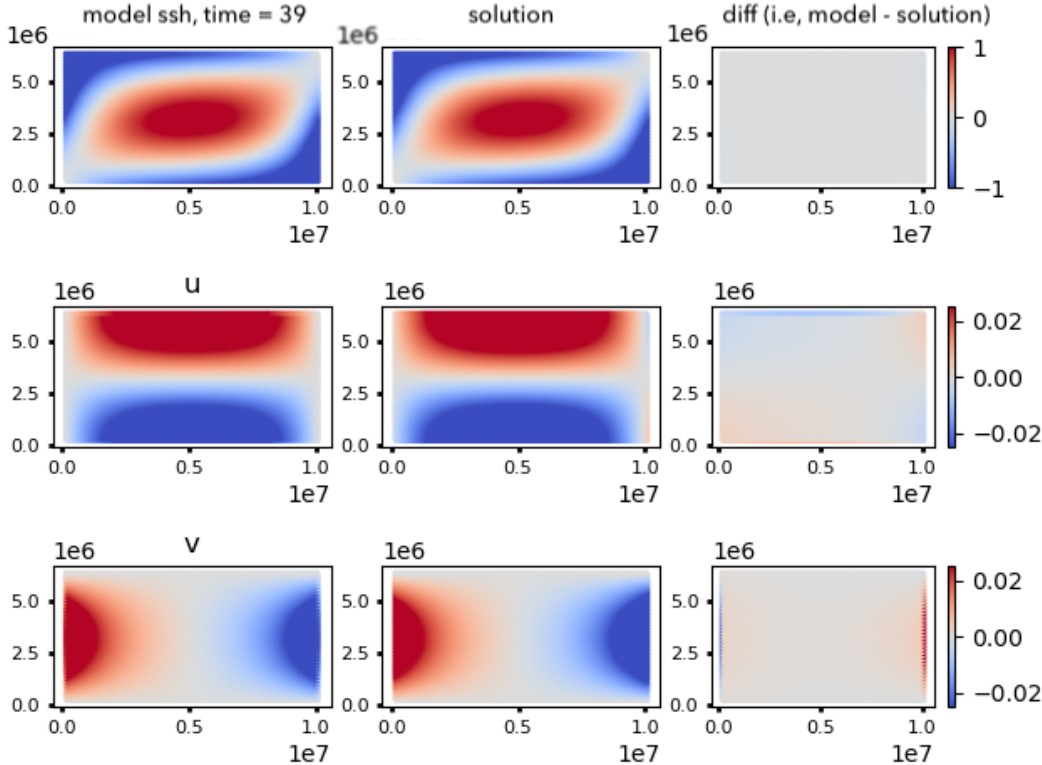

**Figure 2.** Comparison of MPAS-Ocean results against theoretical predictions by Stommel (1948) for the f-plane case: simulation (left column); analytical solutions (middle) and their difference (right) for the sea surface height (m), zonal velocity $u$ and meridional velocity $v$ (m/s). Comparisons were made at 39 days, after the model reached an equilibrium solution. Horizontal axes are $x$ and $y$ (m)

200    Next we check the Stommel solutions for a planar, flat-bottomed ocean basin. Stommel assumed a rectangular ocean with the origin of a Cartesian coordinate system at the southwest corner. The y axis points northward; the x axis is eastward. The shores of the ocean are at $x = 0, \lambda$ and $y = 0, b$. The values of these variables are listed in Table B1 in the Appendix. The ocean is considered as a homogeneous layer of constant depth $D$ when at rest . We observe that MPAS-Ocean simulation results agree very well with the analytical functions derived originally in  (Stommel, 1948).

205    The system is initialized with a sinusoidal wind forcing as shown in the Appendix Fig. B1(b), and we simulate Eq. (1) for different values of coriolis parameter. Below we show simulation results, and also the corresponding analytical solutions, for a typical $f$ plane test case, in which the Coriolis parameter is a constant ($f = 2.5 \times 10^{-4}$). The comparison of the MPAS-Ocean simulation results with the analytical results in shown in Fig. 2. A table with the system parameters are provided in Table B1 in the Appendix.





## 4 Global Tidal Dynamics: Simulation Details

### 4.1 Icosahedral mesh design

The horizontal meshes used throughout this study consist of quasi-uniform Spherical Centroidal Voronoi Tessellations (SCVTs) based on a semi-structured icosahedral decomposition of the sphere. The coarsest grid is composed of 12 pentagonal Voronoi cells (primal grid), and an underlying dual triangular grid (dual grid). Refinements of this mesh are obtained by incremental bisection of the spherical triangles edges, and application of an SCVT optimisation procedure, which iteratively re-positions triangle vertices to lie at the center of mass of their associated Voronoi cells (Ju et al., 2011). The computational grids used in this study were generated using the `JIGSAW` unstructured meshing library (Engwirda, 2017). To facilitate mesh convergence analysis, we employ a set of 4 quasi-uniform, icosahedral-type meshes in this study, with between 7 and 10 levels of structured refinement. The Icosahedron 7 configuration results in approximately 60km global mesh spacing, while the highest resolution Icosahedron 10 configuration leads to approximately 8km global resolution.

### 4.2 Variable-resolution mesh design

In addition to quasi-uniform icosahedral configurations, we also explore the use of unstructured, variable-resolution meshes to enhance the representation of coastal features and regions of high tidal dissipation through selective mesh refinement. Following previous work of Pringle et al. (2021) and consistent with Barton et al. (submitted) and Brus et al. (submitted), we employ a 45-to-5km variable-resolution configuration with mesh spacing $l(\mathbf{x})$ tailored to resolve barotropic wavelengths and sharp bathymetric gradients

$$l_{wav}(\mathbf{x}) = \beta_{wav} T_{M_2} \sqrt{g\tilde{H}}, \tag{10}$$

$$l_{slp}(\mathbf{x}) = \beta_{slp} \frac{2\pi\tilde{H}}{\tilde{\nabla H}}, \tag{11}$$

$$l^*(\mathbf{x}) = \max\left(\min\left(l_{wav}(\mathbf{x}), l_{slp}(\mathbf{x}), l_{max}\right), l_{min}\right), \tag{12}$$

$$l^* \rightarrow |\nabla l| \leq \gamma. \tag{13}$$

Here, $l_{wav}(\mathbf{x})$ and $l_{slp}(\mathbf{x})$ are barotropic tidal length-scale heuristics, with $l_{wav}$ increasing mesh resolution in shallow regions to resolve the wavelength of shallow-water dynamics, and $l_{slp}$ increasing mesh resolution in areas of sharp bathymetry. $\beta_{wav}$ and $\beta_{slp}$ are tunable 'mesh-spacing' parameters, set to $\beta_{wav} = \frac{1}{80}$ and $\beta_{slp} = \frac{1}{4}$ in this study. To control grid-scale noise, $\tilde{H}$ and $\tilde{\nabla H}$ are Gaussian-filtered ($\sigma = \frac{1}{2}$) depths and gradients obtained from the GEBCO2021 bathymetry (GEBCO Compilation Group, 2020). $l^*(\mathbf{x})$ is a combined estimate of mesh spacing throughout the domain, clipping $l_{wav}$, $l_{slp}$ to $l_{min} = 5km$ and $l_{max} = 45km$ globally. To control the gradation of the mesh overall, $l^*(\mathbf{x})$ is limited to enforce a sufficiently slow growth in length scale. We use $\gamma = \frac{1}{8}$ in this study.





### 4.3 TPXO8 and Tidal Evaluation

To assess the accuracy of the implemented of tidal forcing, we compare the amplitude and phase of the tidal constituents
obtained from the MPAS-Ocean simulations against those of data–assimilated TPXO8. We use a number of different meshes
to asses their impact on the accuracy of our results. The meshes used are as follows: (1) a globally uniform icosahedron 7 mesh
with a low resolution of $64km$; (2) a globally uniform icosahedron 8 mesh with a medium resolution $30km$; (3) a globally
uniform icosahedron 9 mesh with a high resolution of $15km$; (4) a globally uniform icosahedron 10 mesh with an ultra-high
resolution of $8km$; (5) a variable resolution mesh with ranging from $45$ to $5km$. A single-layer global ocean configuration
was initialized for each of the above meshes, and without any external wind forcing. An initial ramp–up time of 15 days
was allowed for the tidal potential forcing, after which we observe a steady oscillation in the global mean kinetic energy. We
compare the five major tidal constituents, of which three are semi–diurnal, i.e., $M_2$, $S_2$, $N_2$, and two are diurnal, i.e., $K_1$, $O_1$ as
obtained from our MPAS-Ocean simulations against observational TPXO8 data. The mean harmonic decomposition of each
tidal constituents over a period of 90 days was calculated from the MPAS-Ocean simulations. As a comparison metric, we
choose the global complex root mean square error (RMSE) defined as follows:

$$\text{RMSE} = \left(0.5\left[A_0^2 + A_m^2 - 2A_0 A_m \cdot \cos(\theta_0 - \theta_m)\right]\right)^{\frac{1}{2}} \tag{14}$$

where $A$ is the tidal amplitude, $\theta$ is the phase lag, and the subscripts "o" and "m" refer to the observed and modeled values
respectively. The amplitude RMSE is defined as the limit of RMSE when $\theta_0 = \theta_m$, i.e.,

$$\text{amplitude RMSE} = \left(0.5\left(A_0 - A_m\right)^2\right)^{\frac{1}{2}} \tag{15}$$

$$RMSE_A = \sqrt{\frac{\int\int \text{RMSE}^2 dA}{\int\int dA}} \tag{16}$$

The quantity $RMSE_A$ is weighted by the area $dA$ of each cell. In our study, we perform the tidal analysis calculations for the
$M_2$ constituent of the tide only.

## 5   Results

This section is organized in the following way: in section 5.1 we show the tuning of the linear wave drag coefficient, $\chi$, for
different mesh resolutions. Thereafter, in the same section we show the impact of adding ice shelf cavities on the global tidal
analysis results (in particular the amplitude and phase of the $M_2$ tidal constituent) on the different meshes. In section 5.2 we
show the impact of adding inline SAL (see Barton et al. (submitted) for details) to the MPAS-Ocean simulations on a variable-
resolution mesh. In Section 5.3 we show comparisons of MPAS-Ocean simulation results against data obtained from tide gauge
measurements. In Section 5.4 we compare the results from the different MPAS-Ocean meshes in barchart plots. Finally, we
wrap up our paper with some discussions and conclusions in Section 6.



| Run | $\chi$ | minimum depth | Ice shelf | SAL | M2 RMSE (global)[m] | M2RMSE deep [m] | M2RMSE$_R$[m] | M2RMSE$_{SO}$[m] | RMSE$_{TG}$ | RMSE$_{TGA}$ |
|---|---|---|---|---|---|---|---|---|---|---|
| Icos7$_n$ (≈63km) | 1.8 | 8m | × | scalar | 0.14 | 0.12 | 0.125 | 0.144 | | |
| Icos7$_{is}$ (≈63km) | 1.8 | | ✓ | scalar | 0.135 | 0.125 | 0.12 | 0.01 | 0.12 | 0.115 |
| Icos8$_n$ (≈30km) | 1.44 | 8m | × | scalar | 0.123 | 0.11 | 0.11 | 0.157 | 0.105 | 0.17 |
| Icos8$_{is}$ (≈30km) | 1.44 | | ✓ | scalar | 0.115 | 0.1 | 0.105 | 0.9 | 0.10 | 0.11 |
| Icos9$_n$ (≈15km) | 1.44 | 8m | × | scalar | 0.11 | 0.09 | 0.09 | 0.161 | 0.095 | 0.18 |
| Icos9$_{is}$ (≈15km) | 1.08 | | ✓ | scalar | 0.101 | 0.083 | 0.083 | 0.083 | 0.09 | 0.0815 |
| Icos10$_n$ (8km) | 1.44 | 8m | × | scalar | 0.11 | 0.102 | 0.10 | 0.18 | 0.103 | 0.18 |
| Icos10$_{is}$ (8km) | 1.08 | 8m | ✓ | scalar | 0.093 | 0.088 | 0.088 | 0.076 | 0.1 | 0.085 |
| VR$_n$ (45–5km) | 0.54 | 0.5m | × | inline | 0.11 | 0.053 | 0.053 | 0.12 | 0.053 | 0.12 |
| VR$_{is}$ (45–5km) | 0.54 | 0.5m | ✓ | inline | 0.071 | 0.047 | 0.044 | 0.051 | 0.053 | 0.06 |

**Table 1.** RMSE values of the M$_2$ tidal constituent, calculated for different simulations (Icos7$_n$–VR$_{is}$). M2RMSE (global) refers to the global RMSE value calculated without any ocean depth restrictions; M2RMSE deep refers to the global RMSE value calculated over ocean depths $> 1000m$. M2RMSE$_R$ refers to RMSE values calculated between $66°N$ and $66°S$, and for ocean depths $> 1000m$. M2RMSE$_{SO}$ refers to the RMSE values calculated over the Southern Ocean only, i.e., for latitudes south of $60°S$; RMSE$_{TG}$ refers to RMSE values obtained from the "Truth Pelagic" tide gauge database; RMSE$_{TGA}$ refers to TMSE values obtained from the UHSLC tide gauge data base along the Antarctic coastline.

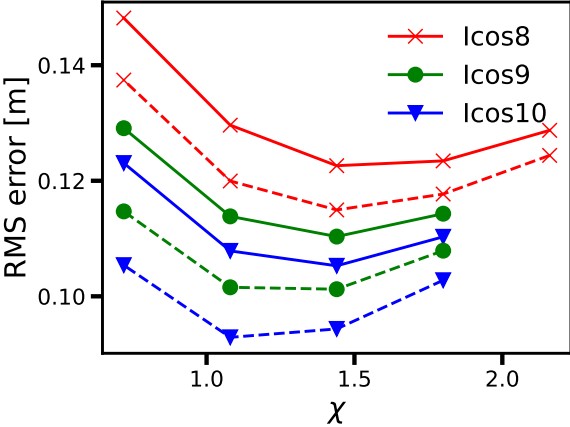

**Figure 3.** Tuning of the topographic wave drag coefficient $\chi$ for runs Icos8$_n$ (solid red with crosses) and Icos8$_{is}$ (dashed red with crosses), Icos9$_n$ (solid green with circles) and Icos9$_{is}$ (dashed green with circles); Icos10$_n$ (solid blue with triangles) and Icos10$_{is}$ (dashed blue with triangles). See Table 1 for reference.

## 5.1 TPXO8 comparisons and wave drag tuning

Due to the presence of a tunable parameter, $\chi$, in the internal wave drag formulation, we carry out a series of simulations on each mesh configuration (Icosahedron 8, 9, and 10) to find the optimum $\chi$ which gives the best agreement against TPXO8 data, i.e., the lowest value of RMSE$_A$ (see Section 4.3 for details). These simulations were carried out with and without adding ice





shelf cavities. From these runs, we conclude that the optimum $\chi$ lies between $0.5$ and $2.5$ (see Fig. 3). Icosahedron $N$ mesh is henceforth referred to as $\text{Icos}N_{is}$ and $\text{Icos}N_n$ for cases with and without ice shelf respectively (here $N = 7, 8, 9, 10$). Note that the optimum value of $\chi$ for $\text{Icos}9_{is}$ ($\chi = 1.08$) and $\text{Icos}9_n$ ($\chi = 1.44$) are different. Similarly for the $\text{Icos}10_{is}$ and $\text{Icos}10_n$ meshes. In particular, the optimum value of $\chi$ is lower when there are ice shelf cavities in the simulation. This fact suggests that ice shelf cavities are responsible for the extra dissipation arising from a lower wave drag in the ocean. The lowest error

obtained from these icosahedron mesh simulations is $0.088m$ (`Icos10`$_{is}$). It must be noted that we have used a scalar SAL term in these simulations.

The effect of tuning $\chi$ on the spatially varying RMSE of the $\text{M}_2$ constituent, for $\text{Icos}8_{n,is}$, $\text{Icos}9_{n,is}$ and $\text{Icos}10_{n,is}$ are shown in Figs. 4(b)–(g) . Evidently, $\text{RMSE}_A$ improves when ice shelf cavities are added in the simulation. In particular, the global $\text{RMSE}_A$ decreases from $8.77cm$ (Figs. 4(b)) to $8.3cm$ (Figs. 4(c)) for the $\text{Icos}8_{n,is}$ meshes respectively. The global

$\text{RMSE}_A$ decreases from $8.0cm$ (Figs. 4(d)) to $7.6cm$ (Figs. 4(e)) for the $\text{Icos}9_{n,is}$ meshes respectively, and similarly from $7.9cm$ (Figs. 4(f)) to $7.4cm$ (Figs. 4(g)) for the $\text{Icos}10_{n,is}$ meshes respectively.

As Fig. 4 shows, we perform two simulations for each mesh resolution, one with no Antarctic ice shelf cavities and one with Antarctic ice shelf cavities. As is apparent from a visual comparison of Figs. 4(f) and (g), the Antarctic ice shelf cavities have the impact of reducing the errors along the Antarctic coastline. For a quantitative estimate of the reduction in errors, in

Fig. 5(a) we show the difference of complex RMSE as obtained from `Icos10`$_n$ and `Icos10`$_{is}$. Fig. 5(b) shows a South Polar projection of the difference in RMSE between `Icos10`$_n$ and `Icos10`$_{is}$.

## 5.2 Effect of ice shelf cavities and SAL

In this subsection we show that adding inline SAL (described in Section 2.4) improves the quality of MPAS-Ocean simulations substantially. In addition, the RMSE values show further improvement when we add antarctic ice shelf cavities to our simula-

tion. We use a variable-resolution horizontal mesh for this part of our work. We show the $\text{M}_2$ RMSE plots for the MPAS-Ocean simulations without and with ice (`VR`$_n$ and `VR`$_{is}$) respectively in Fig. 6. The global deep water RMSE is $0.053m$ for `VR45-5`$_n$ which improves to $0.044m$ (`VR45-5`$_{is}$) when ice shelf cavities are added. The compounded effect of the inline SAL, ice shelf cavity forcing, and the variable-resolution mesh results in the improvement in RMSE values for these two simulations. This is the lowest error achieved using the current stage of development of tides within MPAS-Ocean. As discussed later in this paper,

state-of-art simulations by (Schindelegger et al., 2018; Shihora et al., 2022) have errors of $0.044m$ and $0.034m$, which are very close to the RMSE values in this study. We have shown that carefully–tuned internal tide parameter, inline-SAL, ice-shelves, and locally high-resolution mesh are all necessary to obtain accurate, high quality tides simulations.

## 5.3 Tide Gauge Comparison

To further investigate the impact of ice shelf cavities on tidal dynamics, we compare the results of MPAS-Ocean to tide

gauge datasets including the "ground truth" stations from UHSLC. The locations of these tide gauges are shown in Fig. 7(e). Tidal harmonic data at these stations were consolidated by Pringle (2019). These data including include directly provided tidal harmonics or from using UTIDE (Codiga, 2011) on time level histories. Fig. 7(a)–(d), shows the model vs. tide gauge



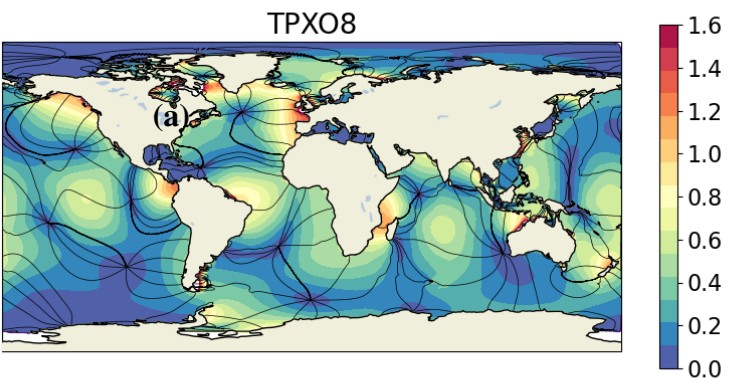

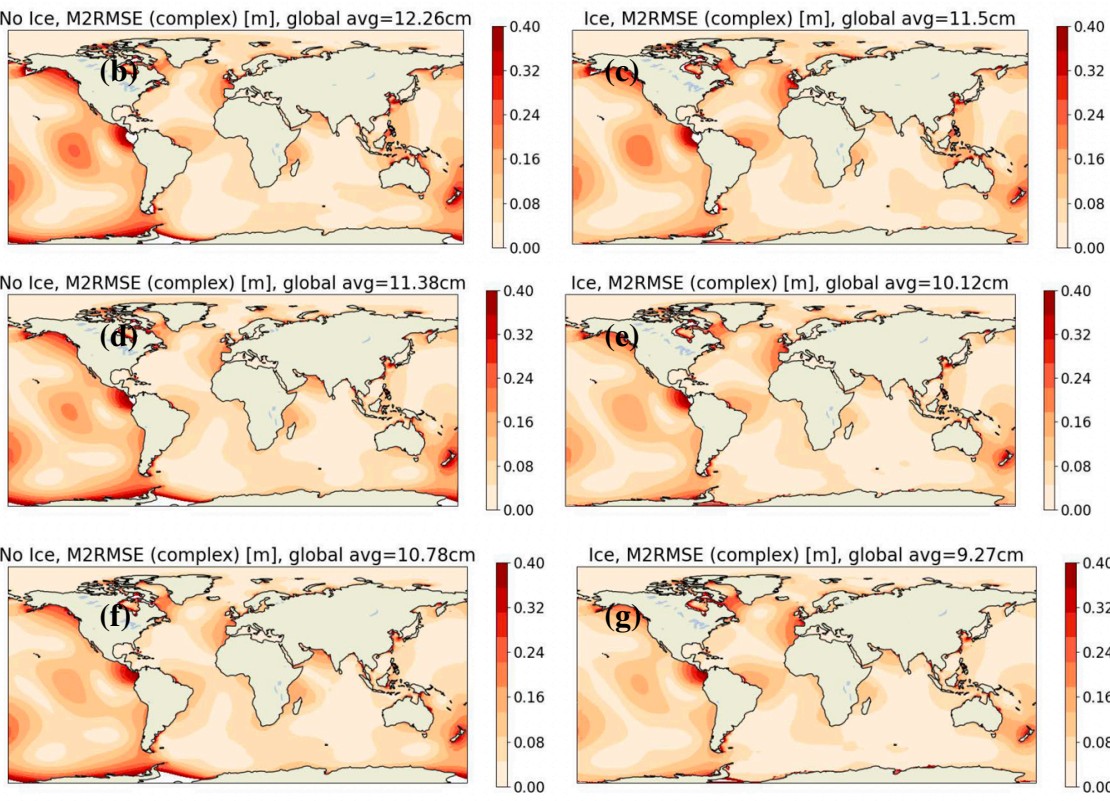

**Figure 4.** Icosahedral global tidal simulation results: (a) reference TPXO8 reDo-analysis mean $M_2$ amplitude, cm, with overlays of $90°$ phase contours; complex RMSE (global, no depth or latitude restrictions) of $M_2$ constituent for (b) `Icos8`$_n$ ; (c) `Icos8`$_{is}$; (d) `Icos9`$_n$; (e) `Icos9`$_{is}$; (f) `Icos10`$_n$; (g) `Icos10`$_{is}$.



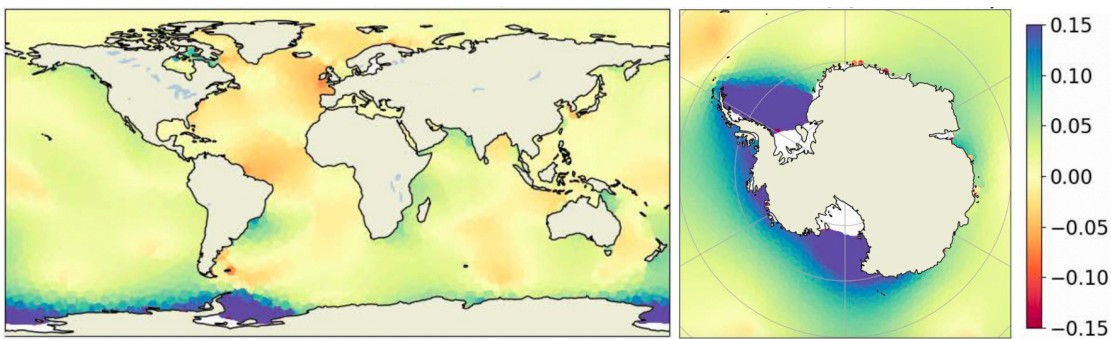

**Figure 5.** Difference of complex RMSE between $Icos10_n$ and $Icos10_{is}$ of (a) global and (b) South Polar projection. See Fig. 4(f) and (g) for reference.

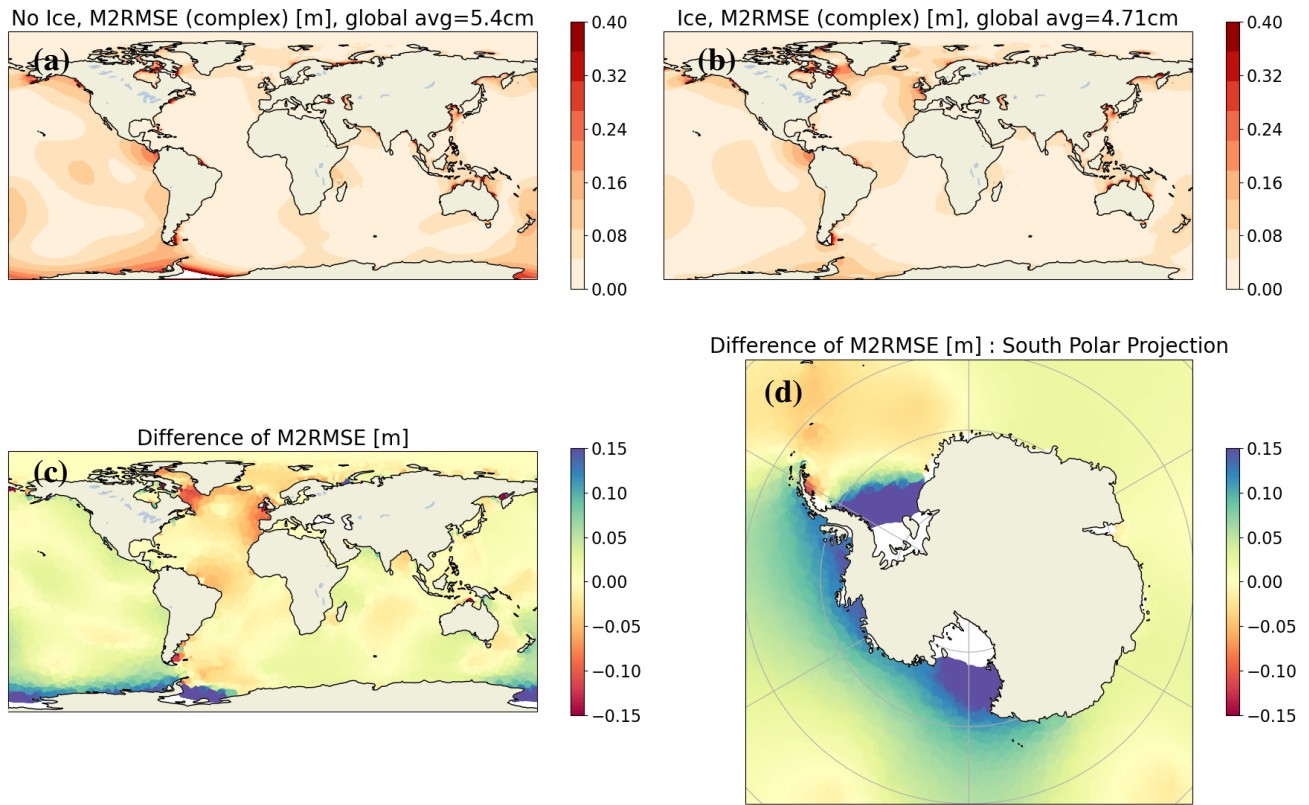

**Figure 6.** $M_2$ RMS error for the variable-resolution mesh and inline SAL (a) without and (b) with ice shelf cavities. The difference between plots (a) and (b) are shown in (c), along with a south polar projection in (d).





amplitudes and phases for runs $\mathtt{VR}_n$ and $\mathtt{VR}_{is}$. These plots include gauges at latitudes south of $60°S$. Figs. 7(a)–(b) show scatter plots of the mean amplitude and phase of the tidal $M_2$ constituent for the run $\mathtt{VR}_n$ (i.e., no ice shelf cavity case). Similarly,

Figs. 7(c)–(d) show scatter plots of the mean amplitude and phase of the tidal $M_2$ constituent for the run $\mathtt{VR}_{is}$ (i.e., with ice shelf cavity case). For the phase data, we shifted the values so that the phase differences were all within 180 degrees. The RMS error when comparing against the 21 UHSLC stations is $0.07$ m for the variable-resolution mesh and $0.085$ m for the 8km quasi-uniform mesh (Table 1), which is consistent with the results seen from the TPXO8 comparison of $0.06$ m and $0.088$m, respectively, for the southern ocean.

We show values of correlation coefficient or the coefficient of determination $R^2$ defined as follows:

$$R^2 = \frac{n \sum xy - (\sum x)(\sum y)}{\left[\sqrt{n \sum x^2 - (\sum x)^2}\right]\left[n \sum y^2 - (\sum y)^2\right]} \tag{17}$$

Here $x$ denotes $M_2$ amplitude (in Fig. 7(a), (c)) and $M_2$ phase (in Fig. 7(b), (d)) from MPAS-Ocean simulation data. $y$ denotes $M_2$ amplitude (in Fig. 7(a), (c)) and $M_2$ phase (in Fig. 7(b), (d)) from tide gauge measurements. When there is no ice shelf in the simulation, the $R^2$ values of the amplitude and phases are $0.8$ and $0.83$ respectively. Including ice shelf cavities increase the

$R^2$ values to $0.955$ and $0.96$ respectively. The ice shelf cavities, although static, provide an accurate boundary condition along the Antarctic coastline, and thus the tidal errors reduce appreciably in the southern ocean. The locations of the stations from which we record the tide gauge data are shown in Fig. 7(e). Fig. 7(e) shows a South Polar projection of the southern ocean, in particular, ocean between latitudes $-90°$ and $-60°$.

## 5.4 Summary of all runs

Finally, Fig. 8 shows three barchart plots exhibiting how ice shelf cavity forcing improves the RMSE in the different regions of the global ocean. In particular, the barcharts are in three categories: (a) the global ocean; (b) the Southern Ocean only (latitudes south of $60°S$); (c) ocean for latitudes north of $60°S$. Although the RMSE shows improvement with ice shelf cavities for all three categories, the strongest improvement is observed for the Southern Ocean (i.e., latitudes south of $60°S$. This result further confirms that Antarctic Ice Shelf cavities are indeed required to accurately capture tides in an Earth System Model.

The lowest error achieved is on the $45km$ to $5km$ variable resolutions mesh, with inline SAL and iceshelf cavities ($\mathtt{VR}_{is}$ with a deep $M_2$ RMS error of $0.044m$. As a point of comparison, Schindelegger et al. (2018) and Shihora et al. (2022) both included full inline SAL calculations into a barotropic tide model and found deep ocean $M_2$ RMS errors of $0.044m$ and $0.034m$, respectively. $M_2$ RMS errors with ADCIRC, were found to be $0.0287m$ by Pringle et al. (2021) further lowered to $0.0194m$ by Blakely et al. (2022). All three of the previous studies used a more optimized wave drag, and the last study

implemented SAL by reading in SAL from a data-assimilated model. Stammer et al. (2014) includes a comparison of errors for various purely hydrodynamical, non-data assimilative models ranging from $0.0525 - 0.0776m$.





**Figure 7.** Tide gauge comparison between the variable-resolution model simulation (horizontal axes) and observations (vertical axes) for $M_2$ amplitude (left column) and $M_2$ phase (right column), where the simulation is without ice shelves ($\mathrm{VR}_n$, top row) and with ice shelves ($\mathrm{VR}_{is}$, bottom row). These tide gauges are near the Antarctic coast and Drake Passage (e).



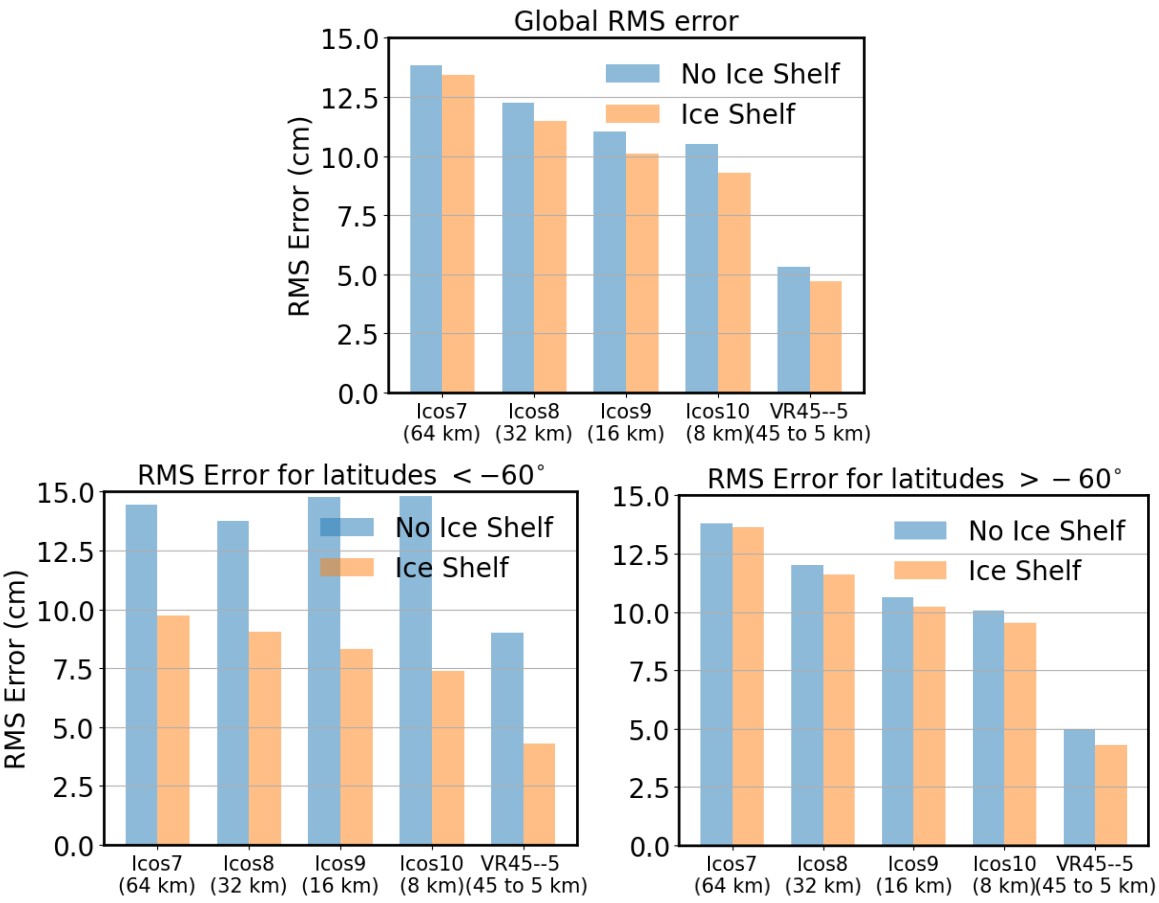

**Figure 8.** Error for the icosahedron 7, 8, 9, and 10 meshes, with and without ice shelf cavities for (a) the global RMSE (b) RMSE at latitudes $< -60°$; and (c) RMSE at latitudes $> -60°$.

## 6 Conclusion

In this paper we have shown the effect of adding tides in a global ocean model, and have explored the effects of drag and ice shelf cavities on global tides. We have computed the tidal analysis on meshes of different resolutions, in particular, the icosahedron meshes and also a variable-resolution mesh. In particular, we use a linear wave drag scheme, and show that a systematic tuning of the wave drag coefficient, $\chi$, results in very accurate tides. We also found that the optimum value of $\chi$ is lower (see Fig. 3) when there are ice shelf cavities in the simulation, indicating the ice shelf cavities provide the extra dissipation. The detailed analysis of simulations with and without ice shelf cavities indicates that introducing static ice shelf cavities reduces the RMS errors appreciably along the Antarctic coastline (see Figs. 4 and 5). Finally, we have shown that using an inline self attraction and loading term, along with ice shelf cavities result in very accurate tides, as shown by comparison





against observational TPXO8 data (see Fig. 6). Our results are further validated by comparison against tide gauge datasets (Fig. 7).

The computation of the full SAL term has been explored in detail in Barton et al. (submitted) and Brus et al. (submitted). In those works, the full SAL has been shown to predict better tides compared to a constant SAL coefficient. Although a detailed
discussion of SAL is outside the scope of this paper, we have shown (using one reference simulation) that inline SAL, along with ice shelf cavity forcing, reduces tidal errors appreciably.

It is clear from this work that ice shelf cavities improve tidal predictions. However, the ice shelf cavities have a fixed geometry in this work. This paper paves the way for future investigations of impacts of varying ice shelf cavity geometries on global tides. Since tides also affect the melting and freezing of ice shelf cavities as shown in Rosier et al. (2014), a two–way interaction
between tides and ice shelf cavities will potentially improve tidal predictions. In this work we have primarily focused on tidal errors in the open ocean near the ice shelf cavities. However, a more detailed analysis of tides and tidal pressures near the grounding line is possible as shown by Begeman et al. (2020). Also, investigations of sea ice dynamics and floating ice shelves are expected to improve tidal errors (Sun et al., 2022; Lei et al., 2017). We would like to conduct more detailed investigations of these factors in a future effort.

*Code availability.* E3SM code is publicly available at https://github.com/E3SM-Project/E3SM/

## Appendix A: Aquaplanet Gravity Wave Test Case

The gravity wave speed on an aquaplanet domain, with no continents, is used to validate the barotropic model. If initialized with a Gaussian hump sea–surface–height test case (see Fig. A1(a)), the Gaussian hump spreads at the speed of gravity waves in the ocean (Figs. A1(b), (c) and (d) represent the Gaussian profiles at times $t = 3.7s, 6.1s, 10s$ respectively). We calculate
the velocity of propagation by tracking the position of a point at the edge of the circular patch in Fig. A1(a), (b), (c), (d). In particular, we track the maximum longitudal coordinate where the sea surface height has a value $0.2m$. The speed of gravity waves would be $c = \sqrt{gH}$, where $g$ is the acceleration due to gravity $= 9.8ms^{-2}$, $H$ is the bottom depth of the ocean (here $1000m$). Thus the gravity wave speed is $\sim 99.8ms^{-1}$. The formula for gravity waves is the solution to the linear wave equation. Since we are running the MPAS-Ocean model with the nonlinear term, the speed will be slightly lower as shown in Fig. B1.
The simulation parameters are shown in Table A1.

## Appendix B: Stommel Basin test case

Next we check the Stommel solutions for a planar, flat–bottomed ocean-basin. Stommel assumed a rectangular ocean with the origin of a cartesian coordinate system at the southwest corner. The y-axis points northward; the x-axis is eastward. The shores of the ocean are at $x = 0, \lambda$ and $y = 0, b$. The ocean is considered as a homogeneous layer of constant depth $D$ when at





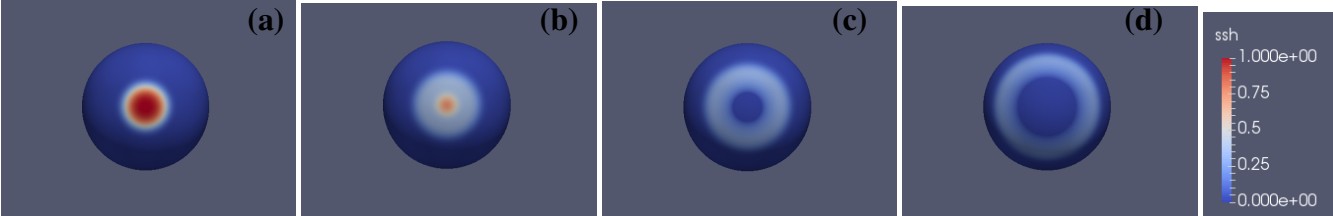

**Figure A1.** Surface gravity wave on a sphere initiated by a Gaussian hump in sea surface height (ssh, m), at (a) $t = 0s$; (b) $t = 3.7s$; (c) $t = 6.1s$ (d) $t = 10s$.

| Run | resolution | Mean speed [$ms^{-1}$] |
|-----|-----------|------------------------|
| AQ1 | 240km | 97.9 |
| AQ2 | 120km | 98.1 |
| AQ3 | 60km | 96.6 |
| AQ4 | 30km | 96.4 |

**Table A1.** Parameters for an Aquaplanet test case simulated using MPAS-Ocean.

rest.Below we show the equations of motion according to Stommel (1948).

$$0 = f(D+h)v - F\cos(\pi y/b) - Ru - g(D+h)\partial h/\partial x \tag{B1}$$

$$0 = f(D+h)u - Rv - g(D+h)\partial h/\partial y \tag{B2}$$

Here $f$ is the Coriolis parameter, $D$ is the bottom-depth (200m in our test case), $h$ is the height above the sea-surface, $u$ is the x component of velocity, $v$ is the y component of velocity, and $R$ is the Rayleigh friction coefficient.

Eqs. (B1)–(B2) are the steady state Navier–Stokes equations. In Stommel (1948), Stommel derived analytical expressions for the sea–surface–height and the horizontal and vertical velocities for a planar domain, and for different values of the Coriolis parameter $f$. The analytical solutions can be exactly verified against the MPAS-Ocean model solutions in the steady state, provided the domain configurations are the same as in Stommel (1948). Thus in MPAS, we choose a rectangular domain with the horizontal edges at $x_{min}, x_{max}$ so that $\lambda = x_{max} - x_{min}$ and the vertical edges at $y_{min}$ and $y_{max}$ so that $b = y_{max} - y_{min}$.

Also, the coefficients in Eqs. (B1)–(B2) can be compared with those of Eq. (1). Rayleigh drag $R$ in Eqs. (B1)–(B2) in $[L][T]^1$, but the rayleigh drag in MPAS is $[T]^{-1}$. Thus $R = 0.02ms^{-1}$ in Eqs. (B1)–(B2) corresponds to $R = \dfrac{0.02}{H} = 0.0001s^{-1}$ in MPAS, where $H = 200m$. The external force is provided by wind stress, which, for the sake of simplicity we consider to be a regular periodic function (Stommel, 1948). The wind force is given by $F = F_0 \cos(\dfrac{\pi y}{b})$. This simple functional form of the external force allows evaluation of analytical solution of Eqs. (B1)–(B2). The wind forcing amplitude is $F_0 = 1Nm^2$ in MPAS,

which when divided by the density $\rho$ gives the amplitude force per unit mass as $\dfrac{F_0}{\rho} = 1e - 3m^2s^{-2}$. In Table B1 we give the different parameters, which are the same for the simulations in this section. We show a visulatization of the wind forcing in the rectangular domain in Fig. B1.




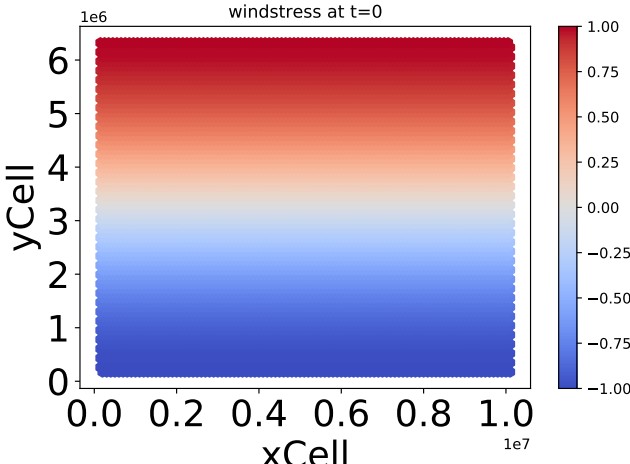

**Figure B1.** Plot of the initial windstress profile for the Stommel Basin test case.

| Run | $N_x$ | $N_y$ | $D(m)$ | $\lambda(km)$ | $b(km)$ | $F_0(m^2 s^{-2})$ | $f(s^{-1})$ | $\beta$ |
|-----|-------|-------|--------|---------------|---------|-------------------|-------------|---------|
| ST1 | 100 | 72 | 200 | 9950000.0 | 6148780.36 | 1.0 | 0 | 0 |
| ST2 | 100 | 72 | 200 | 9950000.0 | 6148780.36 | 1.0 | $2.5 \times 10^{-4}$ | 0 |
| ST3 | 100 | 72 | 200 | 9950000.0 | 6148780.36 | 1.0 | $2.5 \times 10^{-4}$ | $10^{-10}$ |

**Table B1.**

There is a wind forcing at $t = 0$, with the initial condition shown in Fig. B1. We simulate Eq. (1) for different values of Coriolis parameter and list the cases below. We also provide the corresponding analytical solutions and compare the simulation results with the analytical results.

### B1   Non-rotating ocean, $f = 0$

If the Coriolis parameter is 0, the zonal and meridional velocities, and the sea–surface heights have the following analytical solutions in the steady state:

$$u = \gamma(b/\pi)^2 \cos(\pi y/b)(pe^{Ax} + qe^{Bx} - 1) \tag{B3}$$

$$v = -\gamma(b/\pi)^2 \sin(\pi y/b)(pAe^{Ax} + qBe^{Bx}) \tag{B4}$$

$$h(x,y) = -(F/gD)(pe^{Ax}/A + qe^{Bx}/B) -$$
$$(b/\pi)^2(F/gD)(pAe^{Ax} + qBe^{Bx})[\cos(\pi y/b) - 1] \tag{B5}$$

$p = e^{-\pi\lambda/b}, q = 1$

The corresponding comparisons are shown in Fig. 2



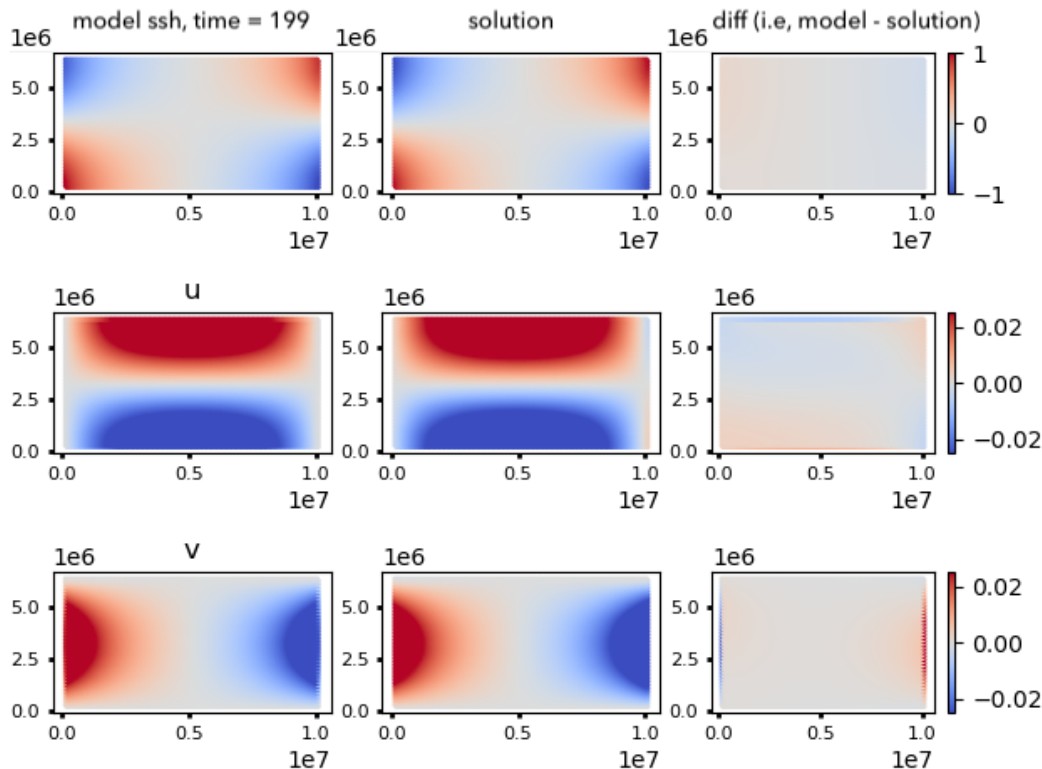

**Figure B2.** Comparison of MPAS-Ocean results against theoretical predictions by Stommel (1948) for the non-rotating case ($f = 0$): simulation (left column); analytical solutions (middle) and their difference (right) for the sea surface height (m), zonal velocity $u$ and meridional velocity $v$ (m/s). Horizontal axes are $x$ and $y$ (m). Comparisons were made after the model reached an equilibrium solution.

## B2   Ocean rotating at a constant angular velocity, i.e., $f = constant$

Here we have $f = 2.5 \times 10^{-4}$ (see Table B1)

The X and Y component of velocities and the sea-surface height have the following expressions.

$$u = \gamma(b/\pi)\cos(\pi y/b)(pe^{Ax} + qe^{Bx} - 1) \tag{B6}$$

$$v = -\gamma(b/\pi)^2\sin(\pi y/b)(pAe^{Ax} + qBe^{Bx}) \tag{B7}$$

$$h(x,y) = -(F/gD)(pe^{Ax}/A + qe^{Bx}/B) - (b/\pi)^2(F/gD)(pAe^{Ax} + qBe^{Bx})[\cos(\pi y/b) - 1] - \frac{f\gamma}{g}(\frac{b}{\pi})^2\sin(\frac{\pi y}{b})(pe^{Ax} + qe^{Bx} - 1) \tag{B8}$$

## B3   Rotating ocean: $f = f_0 + \beta y$

Here, the Coriolis force varies linearly with latitude, i.e., $f_0 = 2.5 \times 10^{-4}$, $\beta = 10^{-10}$.



The X and Y components of velocity, and the sea-surface height expressions are the following:

$$u = \gamma(b/\pi)\cos(\pi y/b)(pe^{Ax} + qe^{Bx} - 1) \tag{B9}$$

$$v = -\gamma(b/\pi)^2 \sin(\pi y/b)(pAe^{Ax} + qBe^{Bx}) \tag{B10}$$

$$
\begin{aligned}
h(x,y) = {} & -(F/gD)(pe^{Ax}/A + qe^{Bx}/B) - \\
& (b/\pi)^2(F/gD)(pAe^{Ax} + qBe^{Bx})[\cos(\pi y/b) - 1] - \\
& (\frac{f\gamma}{g}(\frac{b}{\pi})^2 \sin(\frac{\pi y}{b}) - \frac{\partial f}{\partial y}\frac{\gamma}{g}(\frac{b}{\pi})^3[\cos(\frac{\pi y}{b}) - 1]) \\
& (pe^{Ax} + qe^{Bx} - 1)
\end{aligned}
\tag{B11}
$$

*Author contributions.* NP, KB, MP, DE, SB prepared the paper, designed and implemented the coding upgrades into MPAS-Ocean, designed and performed the experiments, and conducted the tidal analysis. NP led the study, conducted the majority of simulations and analysis, and wrote the first draft of this paper. NP, MP contributed to including the ice shelf cavities, topographic wave drag and bottom drag subroutines within MPAS-Ocean. DE contributed to the coding and development of the JIGSAW software integral to the horizontal mesh creation in this study. KB, SB and MP contributed to the inline SAL calculation software. MP and DE provided project oversight and code review. All authors participated in discussions to formulate the plan, provide feedback, and reviewed and revised the text.

*Competing interests.* The authors declare that they have no conflict of interest.

*Acknowledgements.* This work was supported by the Earth System Model Development program area of the U.S. Department of Energy (DOE), Office of Science, Office of Biological and Environmental Research as part of the multi-program, collaborative Integrated Coastal Modeling (ICoM) project. NP, KB, MP, DE, and BA acknowledge support from PNNL contract DE-AC05-76RL01830. JJW and DW received support from the Joseph and Nona Ahearn endowment at the University of Notre Dame and by the U.S. DOE Grant No. DOE DE-SC0021105. This research used resources of the National Energy Research Scientific Computing Center (NERSC), a U.S. DOE Office of Science User Facility located at Lawrence Berkeley National Laboratory, operated under Contract No. DE-AC02-05CH11231, as well as resources provided by the Los Alamos National Laboratory Institutional Computing Program, which is supported by the U.S. DOE National Nuclear Security Administration under Contract No. 89233218CNA000001



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
