# Peer review of "Barotropic Tides in MPAS-Ocean (E3SM V2): Impact of Ice Shelf Cavities"

_Geoscientific Model Development, 2022_

## Author Response (AR1)

**Cover Letter**

Dr. Juan A. Añel
Geosci. Model Dev. Exec. Editor

5    Re: gmd–2022–188
Barotropic Tides in MPAS-Ocean: Impact of Ice Shelf Cavities
by Nairita Pal, Kristin N.Barton, Mark R. Petersen, Steven R.Brus, Darren Engwirda, Brian K. Arbic, Andrew F. Roberts, Joannes J. Westerink, Damrongsak, Wirasaet

Dear Editors,

10    Thank you very much for sending us the report of the Referees and for giving us the chance to address the concerns of the Referees. We also thank the Referees for taking the time to prepare the report on our paper.

We are pleased to receive the positive review of our manuscript from both Referees. It is encouraging to read that the Referees feel our work on implementing tides in MPAS–Ocean is significant, and will enable more accurate tidal predictions, and hence will facilitate prediction of global climate variability.

15    We give below detailed responses to the questions and comments of the Referees. In addition, we have added the particular version of the Energy Exascale Earth System Model that we use in the title of the manuscript, the main body of the paper, as well as the code availability section, as suggested by the editorial team. The changes in our revised manuscript, which have been made to address the comments of the Referee and the editorial team, are indicated in blue. We have addressed the concerns fully. We hope, therefore, that our paper will now be accepted for publication in Geoscientific Model Development.

20    Thanking you and with our best regards,

Yours sincerely,

Nairita Pal (on behalf of all authors)

The authors are thankful for the reviewers' insightful comments and suggestions that contribute to the improvement of the manuscript. Please find a point-by-point response to each of the remarks made by the reviewers. Modified or added text is marked in blue font in the revised manuscript. All line numbers in the responses refer to the "track changes" version of the document.

**Reviewer 1**

*Thank you for your contribution and the work that you have presented in this manuscript. I have read your paper with great interest. The work presented here falls within the scope of the journal, and the work has been carried out using sound scientific knowledge and principles. Enclosed here are some minor comments*

We thank the reviewer for the positive review of our work. It is truly encouraging to note that the reviewer finds the work interesting and scientific.

The comments are addressed below.

1. *L-16 Precise amplitude and frequency? Needs further explanation*

   Thank you for the comment. The sea surface height oscillations caused by the tidal astronomical forcing has a well–defined mathematical expression, unlike those caused by other factors such as winds, ocean temperature, salinity etc. As we elaborate in Page 5 Section 2.1 (Eqs. 4 and 5), the sea surface height oscillation is given by the equation

   $$\eta_{EQsd,c} = A_c f_c(t_{ref}) L \cos^2(\phi) \cos[\omega_c(t - t_{ref}) + \chi_c(t_{ref}) + \nu_c(t_{ref}) + 2\lambda] \tag{1}$$
   $$\eta_{EQd,c} = A_c f_c(t_{ref}) L \sin(2\phi) \cos[\omega_c(t - t_{ref}) + \chi_c(t_{ref}) + \nu_c(t_{ref}) + \lambda], \tag{2}$$

   However, we understand that the oscillations are not precise due to the effects of ocean circulation, winds, temperature, salinity, etc. Thus in our revised manuscript we replace the word "precise" with "well–defined" (Page 1 L-16).

2. *Fig 2- can the resolution of this figure be improved?*

   Thank you. The resolution of the figure has been improved.

3. *All the equations presented should be centered. Refer to journal guideline*

   Thank you for the comment. Unfortunately, the typesetting of the equations is being done by the journal template automatically, and the equations are indeed slightly off-centered in standard published articles (e.g., this article).

4. *L-69-70 Can be rewritten as it may create confusion for the readers*

   Thank you for pointing out the confusing sentence. We have revised the sentence as follows:

   "The particular scalar drag scheme that we utilize in this study is the Jayne and St. Laurent scheme. This drag acts on a barotropic global ocean model forced with five major tidal constituents: M2, S2, N2, K1, and O1."

   The correction is marked in blue font in our revised manuscript (Page 3).

5. *Fig A1- Is it possible to remove the dark background in this figure*

   The dark background is removed from Fig. A1

6. *The inclusion of ice self-cavities significantly improves the overall tidal prediction; however, how do the effect of altering geometry near the ground line and grounding zone impact the prediction accuracy?*

Thank you for the question. In our work so far, we have not investigated the effect of the grounding line geometry on the prediction accuracy.

*Apart from these minor comments, the overall work that has been presented is significant, well written, and structured and should be considered for publication as the impact of this work will enable accurate tidal prediction, which will facilitate and enable us to predict global climate variability.*

The authors are very happy to read that the reviewer finds the work significant and impactful toward accurate tidal predictions. The reviewer's comments have greatly contributed to the improvement of our manuscript. We hope that our revised manuscript will be considered for publication in this journal.

**Reviewer 2**

*Thank you for your contribution and the work that you have presented in this manuscript. I have read your paper with great interest. The work presented here falls within the scope of the journal, and the work has been carried out using sound scientific knowledge and principles. Enclosed here are some minor comments*

We sincerely thank the referee for the positive feedback of our work. We are indeed happy to read that our paper is an interesting read, and that it is scientifically rigorous. The comments are addressed below.

*L-25 Please provide reference and citations*

Thank you. We have now provided references and citations (page 2 second paragraph of our revised manuscript).

*Fig 1- can Fig 1a be redrawn? it looks like a sketch*

Thank you for pointing this out. The figure has been redrawn (page 7 of our revised manuscript).

*Fig 2 - please improve the figure resolution*

Thank you. The figure resolution is now improved (page 9 of our revised manuscript).

*Fig 4 - please include x and y axis labels*

Thank you. We have now provided the x and y labels (page 13 of our revised manuscript).

*Apart from these minor comments, the overall work that has been presented is significant, well written, and structured and should be considered for publication as the impact of this work will enable accurate tidal prediction, which will facilitate and enable us to predict global climate variability.*

We sincerely thank the reviewer for taking the time to prepare the report for this paper. The comments have led to important improvements of our manuscript. It is truly encouraging to read that the reviewer finds this work impactful towards predictions of climate variability.

---

## Author Response (AR2)

**Cover Letter**

Dr. Rohitash Chandra
Geosci. Model Dev. Topical Editor

Re: gmd–2022–188
Barotropic Tides in MPAS-Ocean: Impact of Ice Shelf Cavities
by Nairita Pal, Kristin N.Barton, Mark R. Petersen, Steven R.Brus, Darren Engwirda, Brian K. Arbic, Andrew F. Roberts, Joannes J. Westerink, Damrongsak, Wirasaet

Dear Editor,

Thank you very much for provisionally accepting our manuscript and for giving us the chance to add the details of the code repository. The full address of the code repository is mentioned in the "Code Availability" section (just before the Appendix section in page 18) of our revised manuscript.

We sincerely thank you for the valuable comments and feedback. We hope that our paper will be accepted for publication in Geoscientific Model Development.

Thanking you and with our best regards,

Yours sincerely,

Nairita Pal (on behalf of all authors)